



# Technical Note: Operational calibration and performance improvement for hydrodynamic models in data-scarce coastal areas

Francisco Rodrigues do Amaral[1], Benoît Camenen[2], Tin Nguyen Trung[3], Tran Anh Tu[4], Thierry Pellarin[1], and Nicolas Gratiot[1,3]

[1]Université Grenoble Alpes, CNRS, INRAE, IRD, Grenoble INP, IGE, Grenoble, France
[2]RiverLy, INRAE, Villeurbanne, France
[3]CARE, Ho Chi Minh City University of Technology (HCMUT), VNU-HCM, Ho Chi Minh City, Vietnam
[4]Vietnam National University-Ho Chi Minh City (VNU-HCM), Thu Duc City, Ho Chi Minh City, Vietnam

**Correspondence:** Francisco Rodrigues do Amaral (francisco.amaral@univ-grenoble-alpes.fr)

**Abstract.**

In this study, we address the challenges posed by data scarcity in hydrodynamic modeling within one of the most vulnerable coastal zones in the world—the Saigon-Dongnai tidal river system in South Vietnam. We investigate calibration strategies for a 1D hydrodynamic model using minimal in-situ data obtained from an existing local monitoring program, which provides 48 hours of measurements per month. To further improve discharge estimation from the 1D model, the coupling of a modified Manning-Strickler (MS) equation is explored. Calibration efforts reveal distinct trends in friction coefficients along the river. The introduction of indirectly measured discharge data significantly improves model performance, particularly for the Saigon River branch. Validation against independent measurements demonstrates promising results, with the coupling of the modified MS equation providing improved discharge estimates. The study underscores the complexities of calibrating hydrodynamic models in data-scarce regions, with recommendations for future modeling endeavors including incorporating more accurate upstream boundary conditions. The long time-series of estimated water level and discharge provided by this study have practical implications for water resource management and decision-making in data-scarce estuarine systems and are provided in open-access for operational use.

## 1 Introduction

Tidal rivers represent intricate systems bridging continental surfaces and the ocean, where water levels and river discharge are profoundly affected by tidal movements. Consequently, the interplay between flooding and tidal constraints on water levels becomes pivotal, particularly in the context of flood protection, pollution management, and climate change adaptation characteristic of these areas. The complex back-and-forth advection coupled with dispersion makes it challenging to predict the propagation of sediment particles or pollutants in such environments.

The Saigon and Dongnai rivers in Vietnam serve as a notable case of a tidal river system, characterized by its flat watershed and, for the Saigon branch, the marginal significance of its net flow compared to tidal influence (Camenen et al., 2021). This river system flows through Ho Chi Minh City (HCMC), one of the most vulnerable megacities globally concerning climate





change impacts (Lossouarn et al., 2016). Research has shown the critical need to understand its hydrodynamics for assessing flood risks (Vachaud et al., 2019), managing saline intrusion (Ngo et al., 2015; Nguyen et al., 2018), and mitigating pollution
(Babut et al., 2019; van Emmerik et al., 2019) and eutrophication (Nguyen et al., 2022).

The limited availability of data for evaluating flow and tidal variations in this region negatively impacts the management of these rivers and their source reservoirs. In April 2024, an intense drought swept through southern Vietnam, with temperatures soaring to nearly 40 degrees Celsius, as reported by several news outlets (Orie, 2024; FranceInfo, 2024). This heatwave significantly impacted irrigation activities and fish populations in the Dongnai region, exemplifying the vulnerability of human
activities to climate change in such areas. Furthermore, it underscores the importance of modeling efforts for this hydrosystem and the necessity of providing free and open-access time-series data of hydraulic variables. Therefore, numerical modeling emerges as a strategic tool for elucidating the behavior of such complex tidal river systems.

The current body of literature contains a few examples of hydrological and hydraulic modeling efforts in the region. For instance, Camenen et al. (2021) estimated the discharge of the Saigon River using two water level measurement points and
a modified Manning-Strickler equation. However, their methodology allows discharge estimation at only one point and relies on continuous water level sensor data. Their discharge estimates are limited to the duration of their measurement campaign from 2017 to 2018. Similarly, Khoi et al. (2022) employed the SWAT model to study the impact of climate change on the Saigon River's discharge. Their model calibration and validation relied on daily discharge data from 1981 to 2000 at four points along the river. Since the study focused on climate impact, the calibration was not as precise as it could have been; the
primary requirement was a reference discharge to compare against future discharge estimates, thereby assessing the impact of climate change. This highlights the scarcity of recent, accurate, high temporal resolution discharge data for this system. This is partly due to the fact that traditional river discharge monitoring methods are labor-intensive, require minimum water depths, or necessitate prolonged measurement periods, making them costly and time-consuming (Eltner et al., 2020). Indeed, river discharge measurements have significantly declined over the past 30 years even in developed countries (Zakharova et al.,
45   2020).

Hydraulic data availability varies significantly among rivers, even in regions with relatively good data coverage, and is particularly scarce in tropical areas (Wood et al., 2023; Scheiber et al., 2023). These limitations can be partially mitigated by using model-generated data (Xu et al., 2022; Heinrich et al., 2023). However, model calibration and validation require in-situ data, which is often lacking. The primary obstacle to modeling efforts for the Ho Chi Minh City region is the very limited
amount of data for calibration and validation. Despite this, Scheiber et al. (2023) managed to set up an urban flood model for HCMC using only open-access satellite data and monthly mean river discharge from reservoir operations. Nonetheless, the model introduced significant uncertainties and limitations inherent to satellite data, aiming to provide preliminary flood maps rather than deterministic conclusions.

This paper aims to i. demonstrate the challenges of operating a 1D hydrodynamic model with minimal data and ii. to
illustrate the utility of a low-cost modeling effort in understanding flow dynamics in a poorly gauged tidal river network. Leveraging the Mage code developed at INRAE Lyon, previously validated on other tidal river systems like the Adour river (Camenen et al., 2022) or the Lower-Seine river (Mendez Rios et al., 2023), the study explores three calibration strategies



for the Saigon-Dongnai river system using scarce in-situ measurements sourced from an already existing local measurement protocol. The calibration approaches put forward are: i. using direct measurements of water level, ii. using discharge data computed from vertical velocity profiles using the velocity index method and iii. using both sources of data. Calibration of friction coefficients poses particular challenges for tidal rivers due to the downstream water level's control over flow dynamics. The River Saigon presents additional difficulties, including the absence of reliable and open-access data on upstream net discharge and the potential impact of extensive canals within the HCMC megalopolis on flow dynamics. Calibration efforts focused on optimizing the Strickler coefficient, $K_s$, by minimizing a loss function comprising water level and discharge relative root mean square errors (rRMSE). Validation against independent measurements was then performed. Additionally, a modified Manning-Strickler (MS) law was coupled with the hydrodynamic model to improve discharge estimation.

In conclusion, this study provides valuable insights and practical implications for water resource management and decision-making in data-scarce estuarine systems. The long time-series of estimated water levels and discharge data are made available in open access, offering a critical resource for both scientific and operational use.

## 2   Materials and Methods

In this study we consider three strategies for calibrating a 1D hydrodynamic model over a complex tidal estuary located in a data-scarce region. The hydrodynamic model under consideration is the Mage code, which will be further presented in sub-section 2.2. The flowchart outlining the methodology is depicted in Figure 1.

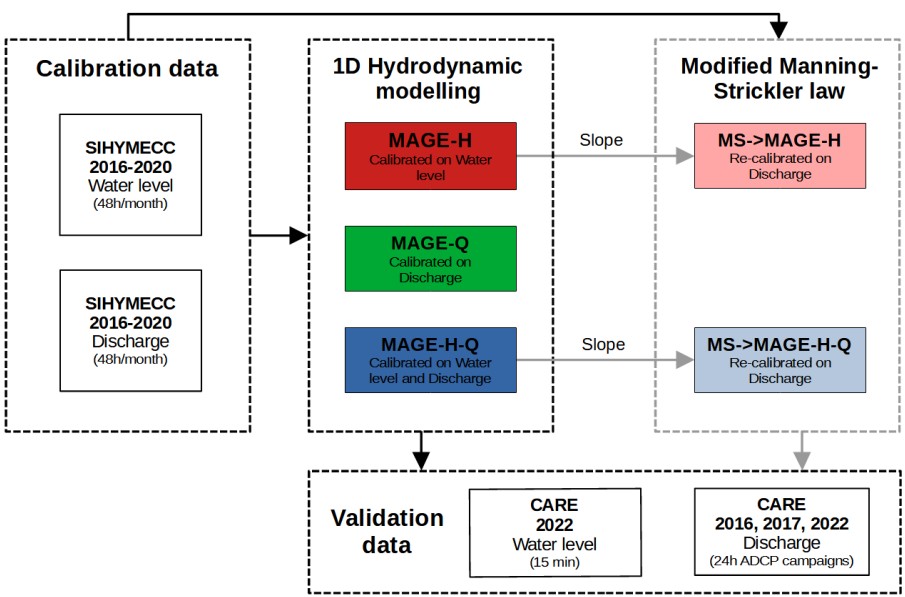

**Figure 1.** Flowchart outlining calibration and validation methods.





The calibration of the Mage model uses minimal in-situ measurements obtained from the Sub-Institute of Hydrometeorology and Climate Change (SIHYMECC), a local Vietnamese agency. The locations for these measurements can be found as white dots in Figure 2. The measurement data, collected using a protocol in place for several years, include hourly river water level and velocity profiles gathered during monthly campaigns of 48-hours from 2016 to 2020. Water level is directly measured with a scale, while discharge is derived from water level and depth-averaged velocity using the velocity index method (Chen et al., 2012). Considering the highly dynamic tidal conditions, the error in discharge estimation is approximately 15% with a minimum error of 150 m³/s (Ruhl and Simpson, 2005).

Due to the quality limitations of discharge data available for calibration, three strategies are employed. Firstly, the Mage model is calibrated solely using water level data and we denote it as MAGE-H. Secondly, Mage is calibrated solely using discharge data, denoted as MAGE-Q. Finally, both water level and discharge data are used for calibration, resulting in MAGE-HQ. In an effort to improve discharge estimation, a modified MS law is coupled with the 1D model. The modification to the classic Manning-Strickler law enables negative slope values and thus, negative water discharge as it is observed in tidally forced rivers (see Section 2.3). This coupling involves feeding slope outputs from MAGE-H and MAGE-HQ into the modified MS law, which is then re-calibrated with discharge data, leading to MS->MAGE-H and MS->MAGE-HQ outputs.

Although coupling a 1D hydrodynamic model with a simplified flow law, such as the Manning-Strickler equation, may initially seem counter-intuitive, it provides essential insights into the dynamics of estuarine rivers. The use of different Strickler coefficients for water levels and discharge, allows for a more detailed calibration of specific components of the system. This approach can enhance the understanding of system behavior by isolating the effects of discharge. It is important to note that recalibrating the MS equation using results from the 1D model aims to refine discharge estimations, despite the fixed water levels that should vary for a modified Strickler coefficient. This refinement can improve operational predictions (as shown in Section 3) for processes such as pollutant advection, nutrient residence times, and salt intrusion. Ultimately, this methodology, when applied with caution, helps to bridge the gap between simplified and detailed modeling approaches, providing a balance between computational efficiency and model accuracy.

The models' validation is conducted using data from the Center of Asian Research on Water (CARE), which includes water level measurements every 15 minutes from October to December 2022 (Rodrigues do Amaral et al., 2023), as well as hourly discharge measurements from four 24-hour Acoustic Doppler Current Profiler (ADCP) campaigns in 2016, 2017, and 2022 (see figure 2, black crosses).

## 2.1 Case study

The Saigon-Dongnai river system, situated in South Vietnam (Figure 2), comprises two main rivers: the River Dongnai and the River Saigon. The River Dongnai originates from Central Vietnam and flows southward through the Tri An reservoir, while the River Saigon originates in Southeastern Cambodia and flows through to the Dau Tieng Reservoir. Downstream of the Dau Tieng reservoir, the River Saigon traverses a highly urbanized area, Ho Chi Minh City (HCMC), which has significant impacts on water quality due to inadequate wastewater treatment (Nguyen et al., 2019). The region features predominantly flat terrain, and the river system is heavily influenced by tides (Camenen et al., 2021).



The instantaneous flow in the Saigon-Dongnai river system can fluctuate between -2000 m³/s to +2500 m³/s, with the net discharge typically remaining below 100 m³/s and occasionally reaching up to 300 m³/s during peak flow seasons. The River Saigon, crossing through HCMC, is interlinked with numerous urban canals and serves as a tributary to the River Dongnai. In contrast, the Dongnai river exhibits substantially higher net discharge values, typically an order of magnitude greater, with monthly averages ranging from 200 m³/s during the dry season to 1200 m³/s during the rainy season (see Figure 4).

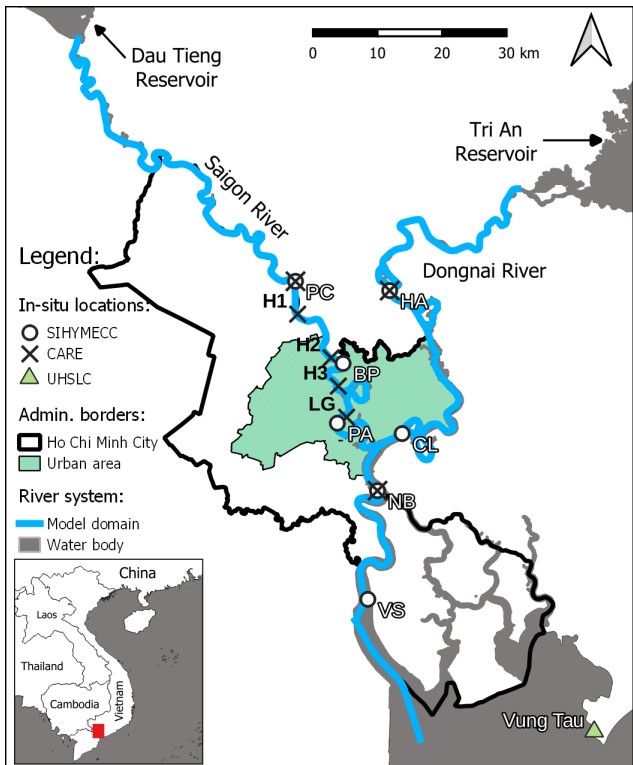

**Figure 2.** Map of the study area. The estuary water bodies (grey) and the Mage model domain (blue) with the Dongnai river (right) and its Saigon river branch (left) can be seen. Locations of river measurements from the Sub-Institute of Hydrometeorology and Climate Change (SYHIMECC, white dots) and from the Center of Asian Research on Water (CARE, black crosses) are also shown. The Saigon river presents six measurement locations namely, Phu Cuong (PC), Hobo 1 (H1), Hobo 2 (H2), Binh Phuoc (BP), Hobo 3 (H3), La Garden (LG) and Phu An (PA). The Dongnai river presents four measurement locations namely, Hoa An (HA), Cat Lai (CL), Nha Be (NB) and Vam Sat (VS). The Vung Tau tidal gauge from the University of Hawaii Sea Level Center (UHSLC) is depicted as a green triangle. The Ho Chi Minh City area and the heavily populated urban center are depicted by a black line and a green polygon, respectively. The area shown in the larger map is represented by the red box in the overview map.



## 2.2 1D Hydrodynamic modelling: Mage

The model for the Saigon-Dongnai river system was constructed utilizing the PamHyr interface presently written in Java and
soon to be accessible as a Python interface (Rouby et al., 2023). This interface offers capabilities for editing network topology,
geometry (including cross-sections and mesh), hydraulic conditions (such as friction coefficients and boundary conditions),
setting numerical parameters, and visualizing primary results. Subsequently, numerical modeling was conducted using the
Mage code (Souhar and Faure, 2009).

The Mage code solves the 1D Barré-de-Saint-Venant equations, comprising the mass conservation equation (Eq. 1) and
momentum conservation equation (Eq. 2):

$$\frac{\partial A_w}{\partial t} + \frac{\partial Q}{\partial x} = q_{lat}, \tag{1}$$

$$\frac{\partial Q}{\partial t} + \frac{\partial}{\partial x}\left(\beta \cdot \frac{Q^2}{A_w}\right) + g \cdot A_w \cdot \frac{\partial Z}{\partial x} = -g\frac{Q \cdot |Q|}{K_s^2 \cdot A_w \cdot R_h^{4/3}} - g \cdot A_w \cdot J_s + k \cdot q_{lat} \cdot \frac{Q}{A_w}, \tag{2}$$

where $A_w$ represents the wet section, $Q$ the water discharge, $q_{lat}$ a lateral input or output (overflow), $Z$ the water surface
elevation, $K_s$ the Strickler coefficient, $R_h$ the hydraulic radius and $J_s$ the a singular head loss. $k$ is a boolean variable such
that $k = 1$ if $q_{lat} < 0$, and $k = 0$ otherwise. Friction head losses are modeled using the classical Manning-Strickler law. Mage
employs a finite difference method utilizing a Preissman scheme and an iterative approach (Newton-Raphson) for solving the
system of non-linear discrete equations (Equations 1 and 2).

Given the geometry of our system domain (Figure 2), the Mage model requires three boundary conditions: (i) a discharge
time series at the source of the Saigon River branch, namely, the Dau Tieng reservoir; (ii) a discharge time series at the source
of the Dongnai River branch namely, the Tri An reservoir; and (iii) a water level time series at the river mouth. A significant
challenge in modeling arises from the lack of data for the upstream boundary conditions, as tidal influence extends to both
upstream dams. As an initial approximation, we will use mean monthly discharge from the period of 2012-2016 as reported
by Nguyen et al. (2019). The upstream boundary conditions are presented in Figure A1 in the Appendix A. Additionally,
a sensitivity analysis showed that the uncertainties on these upstream boundary conditions have a negligible impact on the
instantaneous flow dynamics in the river system (Camenen et al., 2023). However, if the instantaneous discharge were to be
filtered to obtain the net discharge, we would retrieve the upstream boundary condition. Hence, the model output cannot be
used to access the net discharge of these rivers as this variable is an input to the model.

For the downstream boundary condition, we use data from the Vung Tau tide gauge obtained from the research-quality
dataset available through the Joint Archive for Sea Level of the University of Hawaii Sea Level Center (UHSLC) (Caldwell
et al., 2015).

## 2.3 Manning-Strickler law for operational applications

Given Mage's challenges in accurately predicting both water level and river discharge (Camenen et al., 2023), we opted to
couple this model with a discharge estimation approach that has demonstrated efficacy for different tidal rivers (Camenen





et al., 2017), including for the Saigon River branch (Camenen et al., 2021; Rodrigues do Amaral et al., 2024). Illustrated in
Figure 1, the slope output from Mage feeds into a stage-fall-discharge (SFD) rating curve adapted from the general Manning-
Strickler law (Eq. 3):

$$Q(t) = \text{sign}(S) \cdot K_s \cdot A_w \cdot R_h^{2/3} \cdot \sqrt{|S(t)|}, \tag{3}$$

with $Q$ representing the water discharge [m$^3$s$^{-1}$], $K_s$ the Manning-Strickler coefficient [m$^{1/3}$s$^{-1}$], $R_h = A_w/P_w$ the hydraulic
radius [m], where $A_w$ denotes the wet section [m$^2$] and $P_w$ the wet perimeter [m]. The term $\text{sign}(S)$ equates to the sign of the
slope, $S$, taking on values of +1 or -1. The energy slope, $S$ [-], is assumed to be equal to the water slope and is derived from
the water surface elevation output by Mage around the point where discharge estimation is desired. All variables ($S$, $Aw$, $R_h$)
except $K_s$ are as outputted by the Mage model. However, this equation can only yield a discharge value for a specific point in
the river. This point must be a location where discharge measurements exist, as the equation requires calibration of the Strickler
coefficient, $K_s$. Consequently, this integration of a 1D hydrodynamic model and a simplified flow law only has an interest at
locations where SIHYMECC measurements exist (Figure 2, white dots).

## 2.4  Modelling calibration strategy

The calibration of Strickler coefficients in the Mage model follows a sequential approach, starting from downstream to upstream
in accordance with tide propagation. Initially, the Dongnai River is divided into three reaches: two before its confluence with
the Saigon River and one between the confluence and the Tri An reservoir. Subsequently, the Saigon River is divided into seven
reaches: five within the urban center and two between the city center and the Dau Tieng reservoir. The calibration exclusively
utilizes data from SIHYMECC (Figure 1).

For each reach, a bounded Brent's algorithm (Grund, 1979; Brent, 2013) is employed to minimize a loss function. This
is implemented using Python's minimize_scalar function from the Scipy library (Virtanen et al., 2020). The loss function
(Equation 4) is a weighted sum of the relative root mean square error for water level (rRMSE$_H$) and discharge (rRMSE$_Q$)
across all measurement locations. The loss function $f(K_s)$ is formulated as:

$$f(K_s) = \sum_{i=1}^{n} C_i \cdot \left[ \text{rRMSE}_H^i(K_s) + \text{rRMSE}_Q^i(K_s) \right], \tag{4}$$

$$\text{rRMSE}_H^i(K_s) = \sqrt{\frac{1}{m} \sum_{j=1}^{m} \left[ \frac{H_{\text{true}}^j - H_{\text{model}}^j(K_s)}{H_{\text{true}}^j} \right]^2}, \tag{5}$$

$$\text{rRMSE}_Q^i(K_s) = \sqrt{\frac{1}{m} \sum_{j=1}^{m} \left[ \frac{Q_{\text{true}}^j - Q_{\text{model}}^j(K_s)}{Q_{\text{true}}^j} \right]^2}, \tag{6}$$

where $n = 7$ denotes the number of SIHYMECC measurement locations, $C_i$ represents the weight assigned to each measure-
ment location, and rRMSE$_H^i$ and rRMSE$_Q^i$ are the water level and discharge rRMSE for location $i$, respectively. The superscript
$m$ represents each value in the time-series of water level and discharge. The weights ($C_i$) are constant values between 0 and 1,



with higher importance assigned to measurement locations near the urban city center, as detailed in Table 1. This prioritization ensures that the model performance is given more significance around Ho Chi Minh City (HCMC), which is the focal point for flooding applications and other environmental impacts. By weighting the measurements in this manner, we aim to enhance the model's accuracy in areas that are most critical for urban planning and risk mitigation efforts.

Minimizing Equation 4 yields the optimal $K_s$ value for a given reach within the Mage code.

**Table 1.** Weights assigned to each measurement location for the calibration of the Mage model.

| Location | PC | BP | PA | NB | HA | CL | VS |
|---|---|---|---|---|---|---|---|
| Weight | 0.8 | 1 | 1 | 0.8 | 0.8 | 0.8 | 0.5 |

For the modified MS equation, the Strickler coefficient is calibrated against discharge data using the same algorithm as for the Mage model. However, this calibration is performed separately for each location and focuses solely on discharge data. Therefore, the loss function to be minimized is exclusively Equation 6. The calibration of the modified MS equation constitutes a secondary phase of calibration on discharge, as the slope input is already derived from the calibrated Mage model. It is important to note the the discharge data used in this phase of calibration is the same as the one used for the Mage model calibration. Additionally, it is important to reinforce that the modified MS equation can only be applied pointwise, meaning that discharge output is available only at the exact locations where the equation is calibrated, i.e., locations where discharge data is available, notably at the SIHYMECC locations.

The validation of the models utilizes independent measurements from CARE (Figure 1). Two performance metrics are employed: rRMSE and the coefficient of determination, $R^2$. These metrics are computed as follows:

$$\text{rRMSE} = \sqrt{\frac{1}{m} \sum_{j=1}^{m} \left[ \frac{y_{\text{true}}^j - y_{\text{model}}^j}{y_{\text{true}}^j} \right]^2} \times 100, \tag{7}$$

$$R^2 = 1 - \frac{\sum_{j=1}^{m} (y_{\text{true}}^j - y_{\text{model}}^j)^2}{\sum_{j=1}^{m} (y_{\text{true}}^j - \bar{y}_{\text{true}})^2}. \tag{8}$$

where $y_{\text{true}}$ and $y_{\text{model}}$ are the measurement and the model output of the variable of interest, respectively, i.e. water level or discharge.

## 3 Results

Three calibration strategies were tested for the Mage model: calibration solely based on water level, solely on discharge, or on both parameters simultaneously. For the coupled modified MS equation models, namely MS<-MAGE-H and MS<-MAGE-HQ, calibration was conducted exclusively using discharge data. Calibration results for rRMSE and $R^2$ can be found in Figure B1 in Appendix B. For the validation results, the relative Root Mean Square Error (rRMSE) metric is presented in Tables 3 and



2, while the coefficient of determination ($R^2$) can be referenced in Figures C1 and C2 in Appendix C. The resulting Strickler coefficient values across the river extent are illustrated in Figure 3.

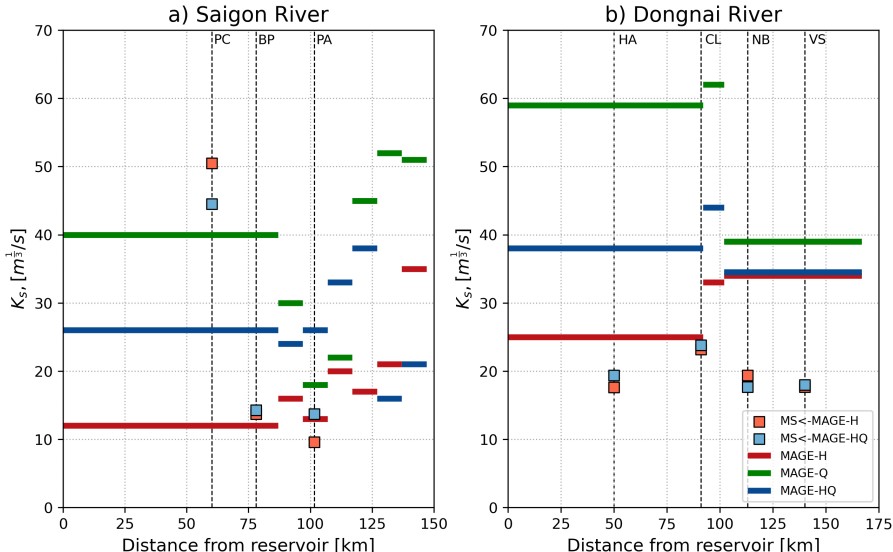

**Figure 3.** Strickler coefficients obtained from the calibration of the Mage model. Solid lines in red, green and blue correspond to MAGE-H, MAGE-Q and MAGE-HQ. The blue and orange squares correspond to the MS<-MAGE-HQ and MS<-MAGE-H at the locations of SIHYMECC measurements.

For the Saigon River, when calibrating the model using water level measurements only (MAGE-H, depicted in red in Figure 3), Strickler coefficients ($K_s$) exhibit an increasing trend as distance from the reservoir increases, with some variability around
the urban city center (100 km to 150 km). When calibrating the model using discharge data only (MAGE-Q, in green in Figure 3), $K_s$ values experience further increments for most parts of the river. Furthermore, introducing discharge data into the calibration process (MAGE-HQ, shown in blue in Figure 3) results in increased $K_s$ values throughout the river, except near the confluence.

Figure 4 presents validation data from the ADCP campaigns plotted against model output. These data, distinct from those
employed in model calibration, stand as fully independent from the model's output (see Figure 1). For the Saigon River, the Strickler coefficients were found to be larger for the MAGE-Q calibration compared to the MAGE-HQ and MAGE-H calibrations. This results in larger discharge amplitudes, as indicated by the red, blue, and green curves in Figure 4. Moreover, the rRMSE between ADCP measurements and model output substantially decreases for the MAGE-Q and MAGE-HQ calibration (refer to Table 2), indicating a significant improvement in model capability despite the lower quality of discharge data
compared to water level measurements. Table 2 also demonstrates that calibrating without utilizing water level data results in an improvement in rRMSE for the Saigon river. However, this is not the case for the Dongnai river as introducing discharge data into the calibration effort provides comparable performance to using only water level measurements. Table 2 shows that




MAGE-HQ calibration yields better results than MAGE-H and MAGE-Q as calibrating using discharge data only leads to overestimation. However, all models are behaving similarly for the asymmetric tide (ADCP campaign of 2016, Figure 4c). In this case the modified MS model does not bring any notable improvements. In terms of net outflow, the modified MS coupling may slightly modify results (Figure 4).

**Table 2.** Validation results for rRMSE [%] between model discharge and ADCP campaigns. The ADCP data was not used for the calibration efforts and thus, model output and validation data are fully independent.

|         | MAGE-H | MAGE-Q | MAGE-HQ | MS<-MAGE-H | MS<-MAGE-HQ |
|---------|--------|--------|---------|------------|-------------|
| PC 2016 | -148   | -62    | -75     | 38         | 31          |
| PC 2017 | -148   | -68    | -78     | 49         | 51          |
| HA 2016 | 31     | 36     | 23      | 43         | 52          |
| HA 2022 | -38    | 47     | 29      | 36         | 40          |

For operational purposes, achieving the most accurate discharge estimation is crucial regardless of the method. Table 2 reveals that coupling the modified MS equation reduces the rRMSE of the Mage model discharge output by half during the symmetric tide of the 2016 campaign and by approximately 20% during the asymmetric tide of the 2017 campaign for the Saigon River. However, errors still range between 31% and 51%, underscoring the challenge of accurately estimating discharge in such dynamic tidal rivers. Conversely, the coupling provides similar or slightly worse results than Mage for the Dongnai River.

Introducing discharge data in the calibration efforts marginally impacts water level output, resulting in slight increases in rRMSE values at most stations (Table 3). However, this calibration approach does not significantly affect the rRMSE values between validation data and model output. The rRMSE difference between MAGE-H and MAGE-Q is consistently below 10%. Additionally, the rRMSE between MAGE-H and MAGE-HQ exhibits minimal differences, always below 5%. Consequently, regardless of the calibration approach, the model effectively reproduces the dynamic nature of water level fluctuations, which is also demonstrated by the $R^2$ values exceeding 0.80 at every measurement station (refer to Figure C1 in Appendix B).

**Table 3.** Validation results for rRMSE [%] between model water level and water level measurements.

|    | MAGE-H | MAGE-Q | MAGE-HQ |
|----|--------|--------|---------|
| H1 | 30     | 38     | 30      |
| H2 | 28     | 32     | 27      |
| H3 | 25     | 27     | 24      |
| LG | 25     | 32     | 30      |
| HA | 31     | 41     | 33      |
| NB | 17     | 21     | 18      |



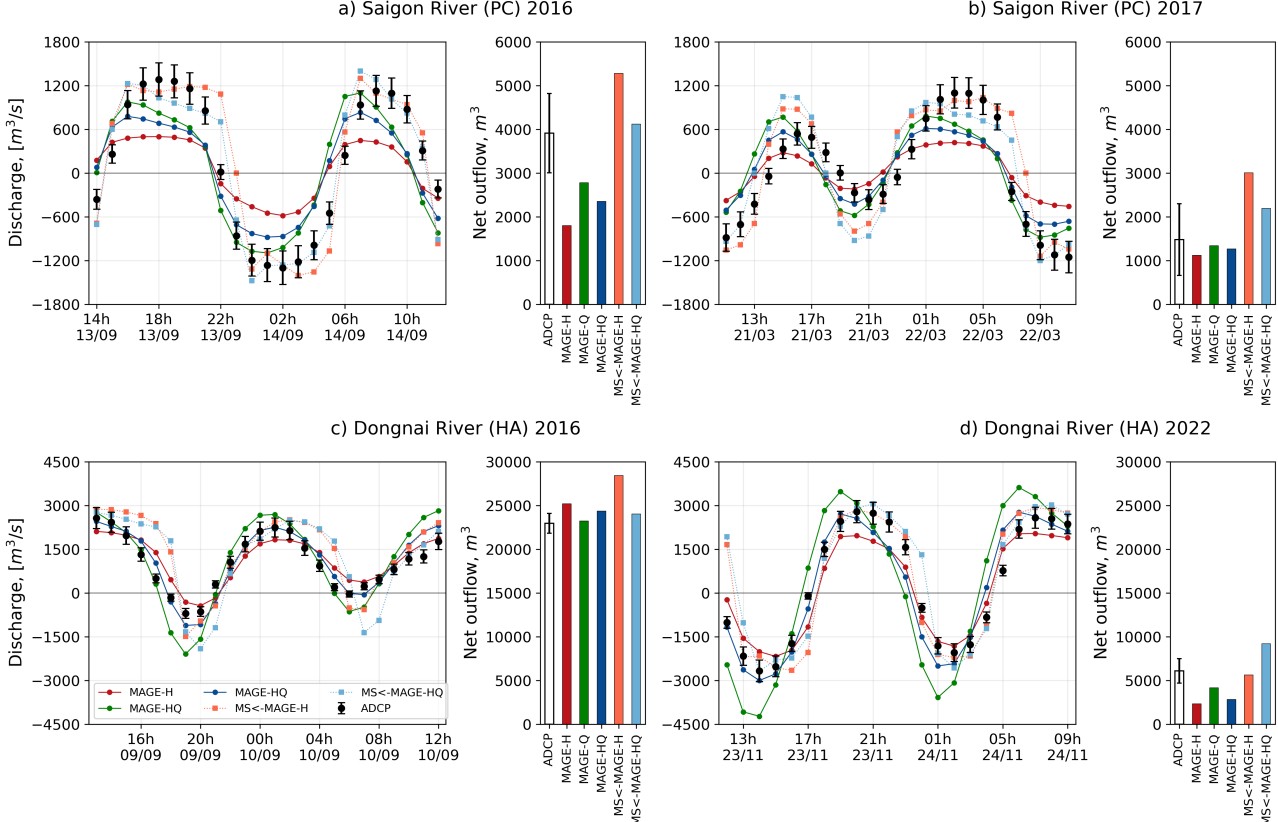

**Figure 4.** Discharge time-series and net outflow comparison between ADCP campaigns and model output. Black dots with uncertainty bar represent the ADCP measurements. Solid lines with dots represent MAGE-H, MAGE-Q and MAGE-HQ in red, green and blue, respectively. Dashed lines with squares represent the MS<-MAGE-H and MS<-MAGE-HQ in orange and blue, respectively. a) and b) Campaigns of September 2016 and March 2017, at the Phu Cuong (PC) location in the Saigon river during a symmetric and asymmetric tide, respectively. c) and d) Campaigns of September 2016 and November 2022, at the Hoa An (HA) location in the Dongnai river during an asymmetric and symmetric tide, respectively.

## 4 Discussion and Conclusions

The calibration strategies employed in this study provide insights into the hydrodynamic modeling of dynamic tidal rivers, particularly the Saigon and Dongnai rivers in Southeast Asia. The three calibration strategies tested for the Mage model, focusing on water level, discharge, or both parameters simultaneously, offer valuable understanding of model behavior under varying calibration conditions with scarce data. Additionally, the coupling of the modified Manning-Strickler (MS) equation with the Mage model presents a novel approach to enhance discharge estimation for operational purposes, albeit with mixed

results across different river systems.





In this tidal-dominated system, the methodology for calibrating friction coefficients involved starting from downstream reaches and then proceeding to calibrate upstream reaches following tidal wave propagation. The model yielded very accurate results for both water levels and discharges in the Dong-Nai River. However, achieving a satisfactory fit for both datasets in the Saigon River proved more challenging. The calibration effort was notably affected by the system's complexity, characterized

by significant tidal discharges and low net discharges originating from the watershed. Notably, the introduction of discharge data from measured vertical velocity profiles significantly impacts model performance, leading to improved accuracy in discharge estimation, especially for the Saigon River. However, the effectiveness of this approach varies, with the Dongnai River demonstrating comparable performance across different calibration strategies.

Modeling efforts are challenging due to the tradeoff between model simplicity for operational use and the accurate rep-

resentation of physical phenomena. For example, the Mage model does not effectively capture tidal wave attenuation, and different calibration strategies result in only negligible improvements. However, when we couple the Manning-Strickler law with a second calibration step, the improvements become significant. This operational trick enhances discharge accuracy, as validation efforts demonstrate. Nonetheless, the one-dimensional nature of the model is not the limiting factor in this case. Using a higher-dimensional model would be impractical, as calibration would require more data, including higher-dimensional

data such as velocity fields, which are hardly available in data-scarce regions like the one under study.

The calibration results reveal distinct trends in Strickler coefficient, $K_s$, values along the river extent, reflecting variations in channel roughness. We observe significant variations in $K_s$ values depending on the calibration strategy and after implementing modified MS coupling. The Strickler coefficient represents the hydraulic roughness of the channel bed, influencing water levels, flow velocity and thus discharge. For instance, a higher Strickler coefficient indicates smoother channel conditions, allowing

for faster flow velocities, while a lower coefficient suggests rougher conditions, resulting in slower velocities. In our study, we observed a range of, $K_s$, values from a very rough value of 10 m$^{1/3}$/s to a smoother 50 m$^{1/3}$/s, indicating substantial variability in channel roughness along the river network. These variations illustrate the challenges of modeling tidally influenced river systems. The Strickler coefficient influences discharge rates during both rising and falling tidal phases, with higher coefficients typically resulting in increased discharges. The Saigon and Dongnai rivers exhibit asymmetrical flow patterns due to tidal

amplitudes, morphology, and coastal features. The impact of the $K_s$ value on discharge varies with the dominant tidal phase and river response to tidal forcing, emphasizing the importance of timing in data collection for calibration and validation. Moreover, the physical behavior of the river may vary at different tidal phases, affecting the Strickler coefficient.

The validation of model outputs against independent measurements underscores the importance of incorporating discharge data for enhancing model capability. Despite the challenges associated with accurately estimating discharge in dynamic tidal

rivers, the Mage model and the MS<-MAGE coupling demonstrate promising results, especially when considering the operational implications of discharge estimation. The impact of calibration efforts on water level output highlights the trade-off between accuracy and simplicity in hydrodynamic modeling. While introducing discharge data may marginally affect water level output, the model retains its ability to capture the dynamic fluctuations in water level, as evidenced by high coefficient of determination (R$^2$) values across measurement stations.



While the current model provides sufficiently good estimates of discharge dynamics within the Saigon-Dongnai system, further refinements are needed, particularly regarding the use of accurately measured upstream boundary conditions. Accurate upstream discharge values from the reservoirs are crucial for model precision. Given that the model is one-dimensional (1D), it inherently requires net discharge as a boundary condition input, preventing the derivation of net discharge directly from the model's results. Consequently, the primary goal of the modeling effort is to obtain more precise instantaneous discharge and

water level estimates, incorporating tidal dynamics throughout the entire river system. This objective has been successfully achieved, and the resulting time series data can provide valuable insights to reservoir managers controlling the river system. Additionally, the modeling effort can be significantly improved if reservoir discharge data are accurately measured and shared with the scientific community. Tidal gauge data, such as those from the Vung Tau station (coastal boundary condition), are already available in near real-time and openly accessible through the University of Hawaii Sea Level Center's website. There-

fore, it is imperative that reservoir discharge data are similarly monitored and made publicly available, as they play a crucial role in improving modeling efforts.

In recent years, there has been growing interest in leveraging advanced statistical and machine learning techniques to enhance the calibration of hydrodynamic models. However, this is not possible in data-scarce regions as these methods rely on large amounts of data. Similarly, machine learning algorithms and AI systems require the leveraging of large datasets for au-

tomating and optimizing the calibration process. On the other hand, Bayesian methods offer a robust framework for calibrating hydrodynamic models (Mendez Rios et al., 2023), thereby improving parameter estimation and predictive accuracy. Integrating these techniques with traditional calibration approaches for 1D models like Mage holds great potential for improving model performance and reducing uncertainty in predictions. Future research endeavors could explore the applicability and effectiveness of such methodologies in the context of dynamic tidal river systems, thereby advancing our understanding and predictive

capabilities in hydrodynamic modeling. In future modelling works, the consideration of canal network impact on the rivers is also recommended. Subsequently, the model could be used to analyze significant flood events. Coupling the model with an advection-dispersion model like AdisTS developed at INRAE (Launay et al., 2019) would also be beneficial in understanding pollutant dispersion in this complex system.

Overall, this study sheds light on the complexities of calibrating hydrodynamic models in data-scarce regions and under-

scores the importance of incorporating discharge data, even if it has important uncertainties associated with it, for enhancing model accuracy. The model outputs have practical implications for water resource management and decision-making in this estuary system and all model output is provided in open-access format in Rodrigues Do Amaral et al. (2024).

*Code and data availability.* The Mage 1D hydrodynamic model is developed by INRAE (French Research Institute for Agriculture, Food and Environment), its source code is written in FORTRAN and can be downloaded here: https://gitlab.irstea.fr/jean-baptiste.faure/mage

[accessed 4. Apr. 2024]. The data and related documentations that support the findings of this study are openly available in DataSuds repository (IRD, France) at https://doi.org/10.23708/KLQMSR. Data reuse is granted under CC-BY license.



## Appendix A: Mage model boundary conditions

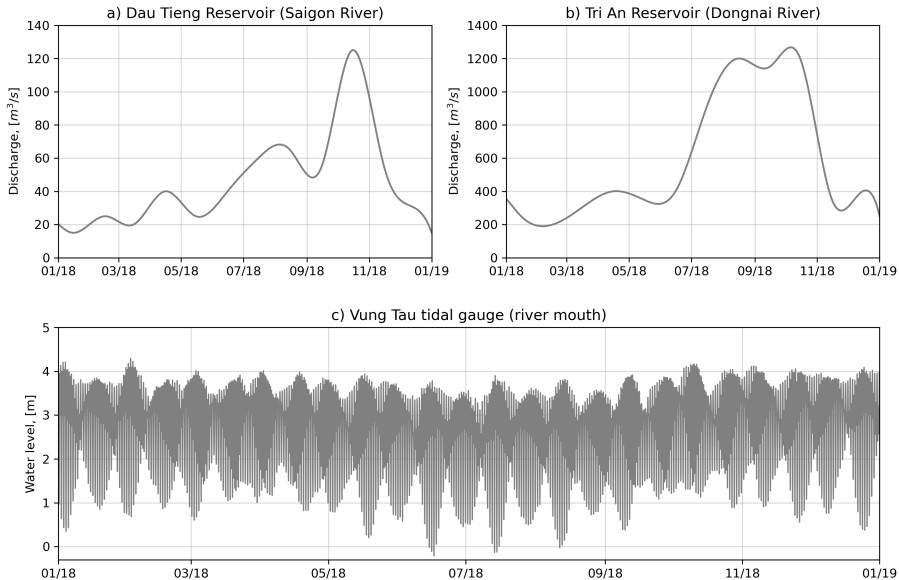

**Figure A1.** Boundary conditions used in Mage model: example for the year 2018. a) Upstream boundary condition for the Saigon river branch: discharge at the Dau Tieng reservoir. b) Upstream boundary condition for the Dongnai river: discharge at the Tri An reservoir. c) Downstream boundary condition at the mouth of the river Dongnai: water level at the Vung Tau tidal gauge.





## Appendix B: Calibration results: rRMSE and $R^2$

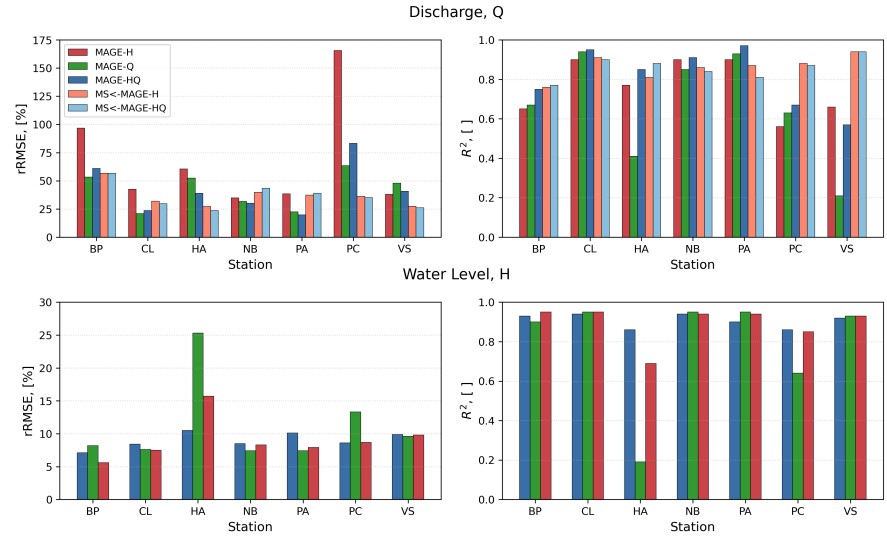

**Figure B1.** Calibration results for the Mage model and for the MS<-MAGE coupling. rRMSE and $R^2$ between model output and calibration data from the SIHYMECC measurement locations are shown. Dark red, green and dark blue bars represent the MAGE-H, MAGE-Q and MAGE-HQ results. Light red and light blue bars represent the MS<-MAGE-H and MS<-MAGE-HQ results.





## Appendix C:  Validation results: $R^2$

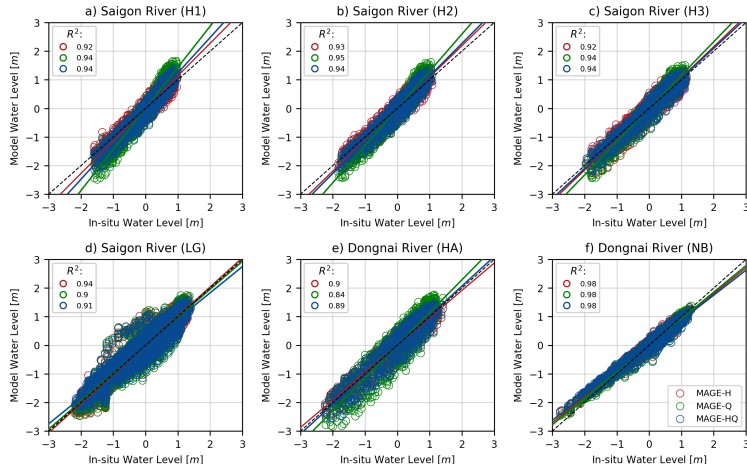

**Figure C1.** Linear regression lines and $R^2$ values between CARE water level measurements and model water level output. The black dashed line represents the $y = x$ line. Red, green and blue circles represent the MAGE-H, MAGE-Q and MAGE-HQ results. Solid lines represent linear regressions for each model with the same color code. a) to d) show results for Saigon locations H1, H2, H3 and LG. e) and f) show results for the locations in the Dongnai river HA and NB, respectively.



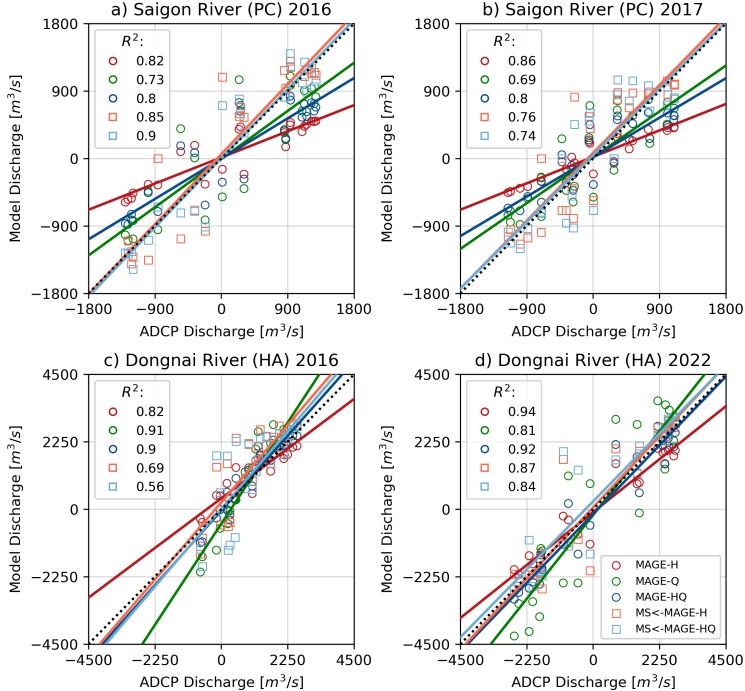

**Figure C2.** Linear regression lines and $R^2$ values between ADCP discharge and model discharge. The black dotted line represents the $y = x$ line. Red, green and blue circles represent the MAGE-H, MAGE-Q and MAGE-HQ results and blue and orange squares the MS<-MAGE-H and MS<-MAGE-HQ results. Solid lines represent linear regressions for each model with the same color code as the circles and squares. a) and b) ADCP campaigns of September 2016 and March 2017, at the Phu Cuong (PC) location in the Saigon river during a symmetric and asymmetric tide, respectively. c) and d) ADCP campaigns of September 2016 and November 2022, at the Hoa An (HA) location in the Dongnai river during an asymmetric and symmetric tide, respectively.

*Author contributions.* Conceptualization, investigation: FRdA and BC; data collection and curation: FRdA and TNT; writing and editing: FRdA; reviewing and supervision: BC, NG, TP and TAT.

*Competing interests.* The authors have declared that no competing interests exist.

*Acknowledgements.* This research was conducted thanks to the financial, technical and human support of the CARE-RESCIF initiative (http://carerescif.hcmut.edu.vn/) within the International Joint Laboratory LECZ-CARE. We would like to thank Dr. Nguyen Truong An for providing the bathymetric data of the River Saigon and River Dongnai. We would also like to acknowledge Mr. Jerard Jardin from the University of Hawaii Sea Level Center for his invaluable work maintaining the Vung Tau tidal gauge.





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
