# Peer review of "Technical Note: Operational calibration and performance improvement for a 1D hydrodynamic model in a data-scarce coastal area"

_EGUsphere, 2024_

## Author Comment (AC1)

**Answer to Reviewer 1**

We sincerely appreciate the thorough evaluation of our manuscript by reviewer 1 and would like to thank them for their comments.

Details outlining these changes, along with other revisions addressing the reviewer's suggestions, are provided below. Below, you will find our responses (in blue) to your feedback **(in black bold).**

**The study involves the development of a coupled 1D and Manning-Strickler (MS) equation. This framework facilitates computationally efficient and accurate estimation of discharges in the data-scarce tidal river system in South Vietnam. The study also discusses the implications of calibrating the framework using discharge and water-level measurements. Though the concept and the calibration approaches are interesting, the paper could benefit from a detailed revision in terms of content, organization, and clarity. The following points need to be addressed.**

**Given that there are several cost-efficient and more accurate modelling approaches, such as the quasi-2D local inertial (Bates et al., 2010; Sridharan et al., 2020) and sub-grid approaches (Neal et al., 2012; Nithila Devi et al., 2024), it is not clear how the proposed method is more computationally efficient and accurate. It would be nice to see a comparison of the proposed approach with a local inertial formulation in terms of accuracy and computational cost.**

Thank you for this comment. We acknowledge that quasi-2D local inertial and sub-grid models have demonstrated advantages in certain applications, particularly for flood extent and inundation modeling. However, the primary objective of our study is not to model urban flooding or floodplain inundation but rather to obtain accurate water level and discharge estimates in a complex, tide-dominated river network with minimal in-situ data. Given this focus, a 1D hydrodynamic model is a priori a more computationally efficient choice.

Additionally, our modeling framework is highly efficient: it requires only five minutes to simulate an entire year, making it well-suited for operational applications and scenario testing. This computational efficiency allows for techniques such as Monte Carlo simulations to explore uncertainties in tidal and riverine processes—something that would be significantly more demanding with higher-dimensional models.

Our 1D approach is computationally more efficient for our specific objective: accurately estimating discharge and water levels in a tide-influenced, data-scarce river system. We acknowledge the potential benefits of comparing different modeling approaches but emphasize that such an analysis would fall beyond the scope of this Technical Note.

**For a longer reach, as used in the study, there may be lateral inflows, and it would be ideal to set up a coupled hydrological and hydraulic modelling system. If the authors are not attempting it, then a valid reason can be provided for the same, or this should be mentioned as a limitation. It is also unclear what the intended use of this modelling framework is – real-time forecasting or long-term continuous simulation. Again, for a longer-duration simulation, it is necessary to couple hydrological and hydraulic models.**

Thank you for this insightful comment. While lateral inflows can be significant over large reaches and in long-term simulations, their impact in our study area is moderated by the dominant tidal

influence, which largely governs flow dynamics. Incorporating lateral inflows would add complexity to the model, reducing its computational efficiency and deviating from our study's goal of developing a streamlined, operational approach. However, it could indeed be of real interest to couple an hydrological model to the hydraulic model. This Technical Note is a first step trying to validate an hydraulic model. Furthermore, the application of an hydrologic-hydraulic model requires better knowledge about water management given that most lateral flows in this system come from irrigation systems and the outflow from the upstream dams. Finally, the water discharge without the effect of tidal movement is 1 order of magnitude lower than the tidally influenced water discharge in these rivers. Hence, lateral inflows are not the main issue affecting our modelling effort.

For these reasons, we do not attempt to couple a hydrological model with our hydraulic modeling system at this stage. However, we acknowledge that in systems with good data coverage and more natural inflows, integrating a hydrological component would be beneficial and this option should still be explored in future work.

We make this clearer in text as follows :

LINES 255-260:

*« While the model provides good estimates of discharge dynamics within the Saigon-Dongnai system, further refinement is needed, particularly with upstream boundary conditions and lateral inflows. Accurate upstream discharge values from the reservoirs are essential for model precision. As a 1D model, it requires net discharge as an input, preventing direct derivation of net discharge from the model's results. **The lack of data of lateral inflows from irrigation and urban canals is seen as an important issue with the modeling efforts presented here. The possibility to couple a hydrological model and incorporating lateral inflows would increase complexity with limited improvement in accuracy.** »*

**The methodology is not clear and difficult to understand. A concise description of the overall framework or the coupling can be provided at the beginning of the methodology section. Is it only the energy slope that has been calculated by the 1D code at the point of interest by the MS equation? If other hydraulic variables are also calculated from the 1D code, it should be mentioned in the methodology flow chart.**

Thank you for your comment. We fully re-structured section 2 - **Materials and Methods** including more details of the MAGE model under sub-section 2.2 - **1D Hydrodynamic modelling : MAGE**.

Indeed, the 1D code provides the energy slope at a point of interest for the MS equation. We will make it clear in text as follows :

LINES 71-72 :

*« Second, to address data limitations and enhance discharge estimation, we integrate a modified Manning-Strickler (MS) law with the 1D model. **This coupling is achieved by using the energy slope computed by the model as input to the MS law.** »*

**The paper should explicitly state the calibration and validation periods used in the proposed method. This is important as the calibration and validation periods have to be distinct (For example, Bhargav et al., 2024). It is unclear from the text which events from 2016, 2017 and 2022 have been used for calibration and validation.**

Thank you for this comment. This has been done in the fully re-structure Methods sections. Indeed, the calibration and validation periods are distint as depicted in Figure 1 : the calibration is done using SIHYMECC data for the period of 2016-2020. This data includes water level data from a water level gauge and discharge data derived from the velocity profiles using the index method. The exact periods are two days every month that do not overlap with CARE data from the same years. The validation is performed using CARE data for 2016, 2017 and 2022. The water level data is obtained from a pressure gauge every 15 min for the year 2022. The discharge data is obtained from four 24-hour ADCP campaigns for the years 2016, 2017 and 2022. The 24h periods of ADCP campaigns do not overlap with calibration data. Explicitly listing all time periods for the data used is impractical, as it involves a large dataset that would require an extensive table beyond the scope of this manuscript. Instead, we focus on explicitly stating that data collection periods do not overlap.

 We make it clearer in the caption of figure 2 as follows :

CAPTION FIGURE 2:

*« Figure 1. Flowchart outlining calibration and validation methods. Calibration data, provided by the Sub-Institute of Hydrometeorology and Climate Change (SIHYMECC), are from non-overlapping time periods separate from the validation data, which were supplied by the Center of Asian Research on Water (CARE). »*

*LINES 69-70:*

*"The performance of each strategy is assessed by validating model outputs against independent datasets from non-overlapping time periods."*

**With the advent of technologies, getting a high-resolution elevation data set would be possible. Whether the surveyed cross-section has been used to construct the 1D model or a high-resolution dataset has been used to represent the bathymetry is not clearly mentioned in the paper.**

Thank you for this comment. The 1D model has been constructed using 83 cross-sections along the Saigon river and 36 cross-sections along the Dongnai river. The bathymetry data was extracted from bathymetry surveys conducted by the (SYHIMECC) in 2016. There is no high resolution dataset of the Saigon and Dongnai bathymetry from the author's knowledge. This and more information on the MAGE model has been added in section 2.2.

This was explicited in text as follows :

LINES 106-108:

*« The river system was constructed with the Mage code using 83 cross-sections along the Saigon river and 36 cross-sections along the Dongnai river. The bathymetry data was extracted from bathymetry surveys conducted by the (SYHIMECC) in 2016 (Nguyen et al. 2021). «*

**Importantly, if the point of this exercise is to have a good simulation of flooding in urban areas, as mentioned in the article, then the ideal choice would be to use a simplified 2D model that can effectively represent the flood inundation dynamics on the complex urban flood plains. The presence of urban infrastructure, such as buildings, roads, streets, etc., governs flood conveyance and distribution. Also, such a claim requires validation using post-flood**

**depth surveys. A coupled 1D and 2D modelling approach can be preferred in this case. Therefore, the authors need to acknowledge the proposed approach's limitations, constraints and advantages. In the introduction, it is also worthwhile to mention whether the focus is only on discharge estimation.**

Thank you for this comment. We would like to clarify that our model is primarily focused on discharge estimation rather than detailed urban flood mapping. While we acknowledge the importance of flood modeling in urban areas, our study is centered on improving the calibration of hydrodynamic models for a river network in data-scarce environments, which is a crucial step toward better understanding flow dynamics in tidally influenced rivers.

In this technical note, we highlight that understanding river hydrodynamics is essential for assessing flood risks, as widely recognized in the field. Our proposed calibration approach enhances the representation of hydraulic processes in poorly gauged tidal river systems, thereby contributing to a broader comprehension of flooding dynamics. However, as stated in the manuscript:

LINES 55-56:

*"This paper aims to (i) demonstrate the challenges of operating a 1D hydrodynamic model with minimal data and (ii) illustrate the utility of a low-cost modeling effort in understanding flow dynamics in a poorly gauged tidal river network."*

Thus, we emphasize that this study does not propose a flood modeling framework but rather a methodological advancement for discharge estimation in complex, data-limited river systems.

**Specific comments**

- **The usage of the term indirectly measured discharge in the abstract is misleading.**

Thank you for this comment. We changed the abstract to read :

LINES :

*« The introduction of discharge data **derived from vertical velocity profiles** significantly improves model performance (…) »*

- **The introduction can include a concise literature review of the existing accurate and computationally efficient hydraulic modelling approaches (not just the ones specific to the region). And briefly discuss how the proposed framework stands out from the existing.**

Thank you for this comment. However, given the format of a Technical Note, an extensive literature review may not be appropriate, as such works prioritize conciseness and focus on a specific technical contribution. Technical Notes are intended to present targeted methodological advancements rather than provide comprehensive state-of-the-art reviews. In this case, the primary focus is on the novel calibration effort in a data-scarce environment, hence the choice to review only scientific literature relevant to the area. Adding a broader discussion on modeling approaches would divert from this objective.

- **Line 27. How is drought modelling relevant here? Are we looking at continuous simulation across all the seasons?**

Thank you for your insightful comment. We agree that drought modeling is not the primary focus of this work. In this particular paragraph, we aim to highlight the significant data gaps in the region, using a recent extreme drought event as an illustration. The impact of this event, particularly on irrigation capabilities and fish populations, could potentially have been mitigated with a deeper understanding of the local hydrosystem. This understanding would include better insights into the river discharge dynamics, impacted by the lack of precipitation.

- **Line 34. Please mention the advantage of the proposed method over Camenen et al. (2021) clearly and why it requires continuous water-level data for calibration.**

Thank you for this comment. Indeed, this sentence is not clear as the calibration in Camenen et al. 2021 does not require continuous water level data. The advantage of this method over Camenen et al. 2021 is the spatial extent : Camenen et al. 2021 obtain river water discharge at a single point in the river Saigon whereas in our study we can provide full coverage of the system, from the mouth of the river system to the upstream dams in both the Saigon and Dongnai branches. The text now reads :

LINES 35-38 :

*"However, their methodology allows discharge estimation at only one point in the river* **corresponding to the location of river water level measurements. As a result, their approach is limited to one specific location in the Saigon branch. In contrast, our study employs a 1D model that captures the full spatial extent of the Saigon-Dongnai river system, providing a more comprehensive representation of its hydrodynamics.**"

- **Line 39. What do you mean by "calibration was not precise?"**

Thank you for your comment. The reduced precision in this calibration can be attributed to two main factors: 1. The calibration was based on discontinuous daily average discharge values, rather than instantaneous discharge data. 2. While this approach is acceptable for climate change studies, where the focus is on relative changes compared to a benchmark (which inherently contains some level of error due to the model's limitations), it becomes more contentious for real-time or near-real-time modeling. In the case of a tidally influenced river like this one, with discharge amplitudes reaching 5,000 m³/s for the Saigon branch and 10,000 m³/s for the Dongnai branch, such a calibration does not capture the necessary temporal resolution for accurate predictions.

- **Line 56. What is Mage code? It is suddenly introduced in the text here.**

Thank you for this comment. The Mage code (Souhar and Faure 2009) is a tool that solves the Saint Venant equations and that was used as basis to the o build the Saigon-Dongnai model. We have postponed its introduction to latter in the text to avoid confusion. The change is as follows :

LINES 56-57:

*« Leveraging a* **1D hydrodynamic model** *developed at INRAE Lyon, previously validated on other tidal river systems (...) »*

- **Lines 60 – 70. These lines look like the conclusion of the study.**

Thank you for this remark. Indeed, it had conclusion like information that has now been removed and the paragraph now reads :

LINES 60-64:

 *« The calibration approaches put forward are: i. using direct measurements of water level, ii. using discharge data computed from vertical velocity profiles using the velocity index method and iii. using both sources of data. Calibration efforts focused on optimizing the Strickler coefficient, Ks, by minimizing a loss function comprising water level and discharge relative root mean square errors (rRMSE). Validation against independent measurements was then performed. Finally, a modified Manning-Strickler (MS) law was coupled with the hydrodynamic model to improve discharge estimation. »*

- **Line 162. Please explain Brent's algorithm in a sentence or two.**

Thank you. We have added the following sentence :

LINES 163-166:

*« Brent's algorithm is a numerical optimization method that combines the robustness of bracketing methods with the efficiency of interpolation techniques to find the minimum of a function. It is particularly effective for univariate optimization problems where the function is continuous but not necessarily smooth or differentiable. »*

- **The results and discussion sections are too wordy. They can be shortened to convey things clearly and concisely.**

Thank you for this comment. Both results and discussion sections have been reduced and improved to be as clear and concise as possible throughout.

**References:**

Camenen, B., Gratiot, N., Cohard, J.-A., Gard, F., Tran, V. Q., Nguyen, A.-T., ...Némery, J. (2021). Monitoring discharge in a tidal river using water level observations: Application to the Saigon River, Vietnam. Sci. Total Environ., 761, 143195. doi: 10.1016/j.scitotenv.2020.143195

Nguyen, A. T., Némery, J., Gratiot, N., Garnier, J., Dao, T. S., Thieu, V., & Laruelle, G. G. (2021). Biogeochemical functioning of an urbanized tropical estuary: Implementing the generic C-GEM (reactive transport) model. Sci. Total Environ., 784, 147261. doi: 10.1016/j.scitotenv.2021.147261

---

## Author Comment (AC2)

**Answer to Reviewer 2**

We sincerely appreciate the thorough evaluation of our manuscript by reviewer 2 and would like to thank them for their comments.

Details outlining these changes, along with other revisions addressing the reviewer's suggestions, are provided below. Below, you will find our responses (in blue) to your feedback **(in black bold).**

**The study presents coupling of a 1D hydrodynamic model with a MS equation for modelling tidally impacted river systems in a data-scarce scenarios. The content of the paper is interesting and novel. However, the paper structure needs to be revised to make the paper more readable and to effectively pass the message to the reader. As such, some of the following points might help improve the paper.**

- **The paper title does not correspond to the content of the paper. For example, 'hydrodynamic models' cover a range of different types of modelling approaches. Please be more specific to reflect what this paper is about.**

Thank you for this comment. We have changed the title to read :

*« Technical Note: Operational calibration and performance improvement for a* **1D hydrodynamic model in a data-scarce coastal area »**

- **Aims and goals of the study are poorly defined and unclear. Please be more explicit in what this study is trying to achieve and what is out side of the scope of this study. Once this is defined it will help to formulate a more specific title for the paper.**

Thank you for this comment. We have made the aims of this clear in text as follows :

LINES 55-56:

*« This paper aims to i. demonstrate the challenges of operating a 1D hydrodynamic model with minimal data and ii. to illustrate the utility of a low-cost modeling effort in understanding flow dynamics in a poorly gauged tidal river network. »*

- **Methodology needs to be improved. More details are needed about the model, input data, model structure and outputs, etc. so that the reader could have more understanding about the model itself. Additionally, more details need to be supplied about how the model was applied to the system. Which characteristics of the system were included and which were disregarded and reasons why.**

Thank you for this comment.  We fully re-structured section 2 - **Materials and Methods** including more details of the MAGE model under sub-section 2.2 - **1D Hydrodynamic modeling : MAGE**. We have provided more information on the input data used in formulating the the 1D model, namely bathymetry and boundary conditions :

LINES 106-108:

*« **The river system was built using 83 cross-sections along the Saigon river and 36 cross-sections along the Dongnai river (Camenen et al., 2023). The bathymetry data was extracted from bathymetry surveys conducted by the SYHIMECC in 2016 (Nguyen et al., 2021).** «*

The primary challenge encountered in this modeling work is the lack of direct measurements of water inputs, not only from the dams but also from tributaries and irrigation canals, which are numerous and significantly influenced by tidal dynamics. These characteristics were left out of the modeling effort. Addressing this limitation in future studies will require improved data collection, particularly on urban canal flows, to enhance model accuracy and better represent the system's hydrodynamics. This was introduced in the text as follows :

LINES 108-110:

*« The primary challenge encountered in this modeling work is the lack of direct measurements of water inputs, not only from the dams but also from tributaries and irrigation canals, which are numerous and significantly influenced by tidal dynamics. These characteristics were left out of the modeling effort. »*

Furthermore , the outputs relevant to this technical note have been made clearer as follows :

LINES 72-74:

*« **This coupling is achieved by using the energy slope computed by the model as input to the MS law.** The MS law undergoes a secondary calibration phase using the same discharge data as the 1D model, after which the discharge outputs are validated against an independent dataset from non-overlapping periods. »*

- **More details about the calibration/validation of the model – especially about the data used, as pointed by the other reviewer.**

Thank you for this comment. Please refer to our answer to the other reviewer's comment on this matter as well as the fully re-structured section 2 – **Materials and Methods**.

- **The results and discussion sections need to be more concise and to the point. At the moment there is too much text, which can be more concise.**

Thank you very much for this comment. This was also pointed out by the first reviewer. Both results and discussion sections have been reduced and improved to be as clear and concise as possible throughout.

---

## Author Response (AR1)

**Answer to Editor**

Dear Dr. Matt,

We sincerely appreciate your valuable feedback and the opportunity to revise our manuscript. We also thank the reviewers for their insightful comments, which have helped us refine and strengthen our work. In this response, we outline the key revisions made to address the concerns raised by you and clarify the contributions of our study.

A major concern was ensuring that our findings extend beyond a site-specific application. We have now explicitly highlighted that the key methodological contribution of this study lies in the calibration approach using a modified Manning-Strickler (MS) equation, which enables accurate discharge estimation in tide-dominated river systems with minimal in-situ data. This calibration method has broader applicability to similar complex hydrodynamic environments where data scarcity is a challenge. Additionally, we emphasize that 1D modeling remains the most efficient choice for estimating water levels and discharges in river networks, particularly where bathymetric data is limited. While 2D models are essential for urban flood modeling, as discussed by Schreiber et al. (2023), their computational cost and data requirements make them less practical for the type of discharge estimation we focus on.

We recognize the need to provide a more thorough justification for the use of a 1D model over a 2D approach. To clarify this, we have expanded the discussion in the introduction and methods section. First, in terms of computational efficiency, a 1D model can simulate a full year in just five minutes, which is significantly faster than 2D models, making it ideal for uncertainty quantification, scenario testing, and practical applications in data-limited settings. Furthermore, 1D models facilitate Monte Carlo methods. As demonstrated in similar tidal systems by Terraz & Mendez-Rio (2023), 1D modeling has been successfully used for computing statistical parameters such as Sobol indices to characterize spatially distributed friction coefficients. Finally, unlike 2D models that focus on floodplain dynamics, our study is centered on accurate discharge and water level estimation, which aligns with the strengths of a 1D approach.

The suggestion to incorporate Markov Chain Monte Carlo (MCMC) or other uncertainty quantification techniques is well taken. However, we clarify that the primary aim of this technical note is to present a novel calibration technique rather than a full uncertainty analysis. We acknowledge the importance of uncertainty quantification and discuss it as a potential avenue for future research, where the validated 1D model can be used in probabilistic framework.

In response to concerns about the clarity of our writing, we have thoroughly revised key sections. In the abstract, we have removed vague statements and provided precise error metrics (rRMSE reductions) to quantify model accuracy. In the introduction, the study's aims have been reframed to focus on overcoming challenges associated with 1D model calibration. In the methods section, we have clarified details regarding calibration and validation periods to improve transparency.

Thank you again for your time and consideration.

Best regards,

The authors

References :

Schreiber et al. (2023). Nat. Hazards Earth Syst. Sci., 23, 2313–2332; https://doi.org/10.5194/nhess-23-2313-2023

Terraz, T & Mendez-Rios, F. (2013). Coupling Mage with Melissa to Compute Ubiquitous Sobol Indices for River Hydraulics. Advances in Hydroinformatics—SimHydro 2023 Volume 1, Ed.: Gourbesville, F. & Caignaert, G., Springer, pp. 25-39